# Prevalence and Psychosocial Correlates of Diabetes Mellitus in South Africa: Results from the South African National Health and Nutrition Examination Survey (SANHANES-1)

**DOI:** 10.3390/ijerph20105798

**Published:** 2023-05-12

**Authors:** Sibusiso Sifunda, Anthony David Mbewu, Musawenkosi Mabaso, Thabang Manyaapelo, Ronel Sewpaul, Justin Winston Morgan, Nigel Walsh Harriman, David R. Williams, Sasiragha Priscilla Reddy

**Affiliations:** 1Public Health, Societies and Belonging, Human Sciences Research Council, Pretoria 0001, South Africa; ssifunda@hsrc.ac.za (S.S.); mmabaso@hsrc.ac.za (M.M.); 2School of Medicine, Sefako Makgatho Health Sciences University, Ga-Rankuwa 0208, South Africa; tonymbewu@gmail.com; 3Social Science Core, Africa Health Research Institute, Somkhele 3925, South Africa; 4Public Health, Societies and Belonging, Human Sciences Research Council, Cape Town 8000, South Africa; rsewpaul@hsrc.ac.za; 5Department of Social and Behavioral Sciences, Harvard T.H. Chan School of Public Health, Boston, MA 02115, USA; jmorgan@g.harvard.edu (J.W.M.); nharriman@hsph.harvard.edu (N.W.H.); dwilliam@hsph.harvard.edu (D.R.W.); 6Department of African and African American Studies, Harvard University, Cambridge, MA 02138, USA; 7Faculty of Health Sciences, Nelson Mandela University, Port Elizabeth 6031, South Africa; sasiragha.reddy@gmail.com; 8The Centre for Critical Research on Race and Identity, University of KwaZulu-Natal, Durban 4041, South Africa

**Keywords:** non-communicable diseases (NCDs), diabetes, psychosocial determinants, psychological distress, epidemiological transition, Black South African, urban, rural, South Africa

## Abstract

In South Africa, there are a limited number of population estimates of the prevalence of diabetes and its association with psychosocial factors. This study investigates the prevalence of diabetes and its psychosocial correlates in both the general South African population and the Black South African subpopulation using data from the SANHANES-1. Diabetes was defined as a hemoglobin A1c (HbA1c) ≥6.5% or currently on diabetes treatment. Multivariate ordinary least squares and logistic regression models were used to determine factors associated with HbA1c and diabetes, respectively. The prevalence of diabetes was significantly higher among participants who identified as Indian, followed by White and Coloured people, and lowest among Black South Africans. General population models indicated that being Indian, older aged, having a family history of diabetes, and being overweight and obese were associated with HbA1c and diabetes, and crowding was inversely associated with HbA1c and diabetes. HbA1c was inversely associated with being White, having higher education, and residing in areas with higher levels of neighborhood crime and alcohol use. Diabetes was positively associated with psychological distress. The study highlights the importance of addressing the risk factors of psychological distress, as well as traditional risk factors and social determinants of diabetes, in the prevention and control of diabetes at individual and population levels.

## 1. Introduction

Diabetes mellitus (DM) is a chronic disease caused mainly by either the lack of production of insulin (in approximately 8% of diabetics due to autoimmune destruction of the insulin-producing beta cells of the pancreas: type 1 diabetes) or the ineffective utilization of insulin produced by the pancreas (in approximately 90% of diabetics due to insulin resistance: type 2 diabetes) [1]. The rapidly increasing global prevalence of diabetes poses a major public health challenge caused by the global epidemic of obesity, nutritional transitions, sedentary lifestyles, and other risk factors for type 2 diabetes [2]. An estimated 9.3% of the total global population (734 million people) currently have diabetes [3]. The prevalence is expected to rise to about 10.4% (822 million people) by 2040. Diabetes is already one of the top 10 causes of death globally and is even higher in high-income and middle-income countries [4].

The prevalence of diabetes mellitus has rapidly increased in South Africa, from 4.5% in 2010 to 12.7% in 2019. Of the 4.58 million people aged 20–79 years who were estimated to have diabetes in South Africa in 2019, 52.4% were undiagnosed [5]. South Africa has the second highest number of people living with type 2 diabetes in sub-Saharan Africa [6]. In South Africa, diabetes is the leading underlying natural cause of death in women and the second highest underlying cause of death for the entire population [7]. Diabetes and its complications are strongly associated with modifiable risk factors and determinants. Previous studies of diabetes in South Africa have focused on the traditional determinants of diabetes and its comorbidities, investigating how socio-demographic factors (socioeconomic status, age, sex, marital status, level of education, income, occupation, social position, and residential area) and behavioral factors (smoking, poor diet, physical inactivity, excess alcohol consumption) impact diabetes prevalence and management [8]. Over the past three decades, since the advent of democracy in the country, increasing household income and urbanization have led to accelerated changes in environmental and social stressors, diet, and physical activity behaviors of South Africans, predisposing them to increased risk for a range of non-communicable diseases (NCDs), including diabetes. This epidemiological transition is evident in the rapidly rising levels of obesity and the increasing prevalence of cardiovascular disease over the past 25 years [9].

The epidemiological transition was a concept first articulated by Omran [10]. It describes how, in societies experiencing increasing modernization, aging, and life expectancy, the national disease profile changes from predominantly communicable diseases to that of NCDs, such as diabetes and cardiovascular disease. Omran’s theory focused on the “complex change in patterns of health and disease; and on the interactions between these patterns and their demographic, economic and sociologic determinants and consequences” [10]. The ecological changes of the epidemiological transition include nutrition transitions as well as urbanization, which are brought about by increasing globalization and socioeconomic development.

South Africa remains a largely unequal society with a Gini coefficient of 0.7, one of the highest in the world. Racial inequities are present in household income, access to services, health care, employment, and geographic location [11]. Socioeconomic status (SES) is a strong predictor of health outcomes among different racial groups [12]. The historical and current racial disparities may influence the prevalence of NCDs such as diabetes in South Africa. Furthermore, within the Black South African subpopulation itself (which comprises almost 80% of the South African population) there are considerable variations in SES, health status, behavioral risk factors for NCDs, and access to health services by urban/rural status. There is a dearth of population-based studies aimed at investigating NCD risk within the Black South African population in South Africa. However, one recent study using national data found that hypertension risk varied by geographic location for Black South Africans, with a lower prevalence of hypertension in rural informal compared to urban formal areas [13]. Thus, an improved understanding of the influence of the geographic, social, economic, and cultural heterogeneity within the Black South African population on diabetes risk is needed.

Diabetes and its complications are also likely associated with non-traditional psychosocial risk factors such as psychological distress (symptoms of depression and anxiety) and social stressors. In turn, these non-traditional risk factors can affect the physical, social, and mental well-being of people living with diabetes [14]. Furthermore, psychosocial factors (emotional and psychological distress, exposure to life stress and early life adversity, and environmental and social stress) influence chronic disease management [15]. However, most prevention and treatment interventions for diabetes have focused on mitigating traditional risk factors. In South Africa, limited research has been conducted concerning non-traditional risk factors for diabetes, including the above psychosocial factors. This is partly due to a lack of population-based data on diabetes and NCDs in the country [4].

The 2012 South African National Health and Nutrition Examination Survey (SANHANES-1) was one of the first population-based surveys designed to, among other goals, assess and profile the burden of NCDs, including diabetes, in South Africa [16]. Furthermore, SANHANES-1 investigated the prevalence and psychosocial correlates of diabetes (such as race, psychological distress, social and environmental stressors, and health risk behaviors) in the South African population using a nationally representative sample. The survey also captured respondents’ geographical location (urban formal, urban informal or tribal, rural formal, and rural informal or farms). Such geographic determinants are especially significant in the Black South African population, who suffered the effects of Apartheid laws such as the Group Areas Act, which confined them to impoverished and socially deprived sectors of the cities and forced removals that dumped “surplus” people in rural slums [17,18,19]. As a consequence, large socioeconomic disparities between races persist to this day, as a legacy of Apartheid and produce large health inequalities between different races. Many of these socioeconomic and health inequalities are a result of the geographic distortions (urban/rural communities; residence in formal/informal settlements) that Apartheid and colonialism produced.

This study seeks to explore the association between socio-demographic characteristics and psychosocial exposures, diabetes, and HbA1c in South Africa. Additionally, the paper considers variations in diabetes risk within the Black South African population by geographic location. In South Africa, geographic location reflects, in large part, historical disparities and Apartheid spatial development policies. It is important to understand the extent to which the emerging and increasing prevalence of diabetes varies for population groups that historically reported a very low prevalence of diabetes, especially for Black South Africans in rural communities.

## 2. Materials and Methods

### 2.1. Data Source

This study used secondary data from the 2012 South African National Health and Nutrition Examination Survey (SANHANES-1). This nationally representative population-based cross-sectional household survey was conducted using a multi-stage disproportionate, stratified cluster sampling approach described in detail elsewhere [16]. Briefly, a total of 1000 census enumeration areas (EAs) from the 2001 population census stratified by province and locality type, including race in urban areas, were used as a basis for the sampling of households. A sample of 20 visiting points was randomly selected from the EAs and this yielded a sample of 10,000 households, of which 8166 were valid, occupied households. Of these households, 6306 (77.2%) agreed to participate in the survey. This resulted in a total of 27,580 eligible individuals (household members), of which 92.6% participated in the survey.

All persons living in occupied households were eligible to participate in the survey. The survey examined socio-demographic information, self-reported family history of NCDs, and self-reported health conditions combined with a physical examination, clinical tests, and selected blood sampling tests for disease biomarkers. Of the 25,532 (92.6%) individuals who completed the interviews, 12,025 (43.6%) underwent a medical examination, and 8078 (29.3%) provided a blood sample for biomarker analysis [16]. Only 4598 individuals 15 years and older with data on diagnosed diabetes were included in this analysis.

### 2.2. Primary Outcome

The two primary outcome variables were (1) HbA1c, a continuous outcome variable and (2) the presence or absence of diabetes, a binary outcome variable. In this dichotomous variable, a diagnosis of diabetes was based on the World Health Organization and American Diabetes Association criteria of an HbA1c higher than 6.5% and/or currently taking medication for diabetes [20,21].

### 2.3. Explanatory Variables

#### 2.3.1. Socio-Demographic Variables

Socio-demographic variables included age, sex (male and female), race (reported as per Statistics South Africa’s standard population groups: Black South African, Coloured, White, and Indian), educational status (no formal schooling, grades 8–12, higher education), geographic locale (formal urban, informal urban, formal rural, or informal rural), annual per capita household income in Rands (<5000; 5000–9999; 10,000–24,999; 25,000–49,999; ≥50,000; intervals that correspond approximately to the expected exponential distribution of income), an asset-based wealth index (constructed by summing various household amenities and asset ownership to compute five quintiles: 1st lowest, 2nd lower, 3rd middle, 4th higher, and 5th highest) representing a continuum of household SES from the poorest to the least poor. The race categories Black South African includes people who identify as indigenous African, Coloured includes people who identify as having mixed ancestry, White includes people who identify as having European ancestry, and Indian includes people who identify as having Indian subcontinent ancestry [22].

#### 2.3.2. Health-Related Variables

Health indicators included family history (FH) of diabetes, body mass index (BMI) (underweight < 18.5 kg/m^2^, normal weight 18.5–24.9 kg/m^2^, overweight 25–29.9 kg/m^2^, obese ≥ 30 kg/m^2^), inactive lifestyle (<2000 metabolic equivalent of task (MET) minutes per week), low fruit and vegetable intake score (≤2, the high score is 8), high sugar intake score (≥5, high score is 8), high fat intake (≥11, the high score is 20), and high alcohol use (by the AUDIT-C, scores ≥ 4 for men and ≥3 for women indicate high alcohol use [23]. Fruit and vegetable, sugar, and fat intake scores were computed by the sum scores of the four, four, and ten questions, respectively, on the frequency of past-week consumption of each of these foods, and the first and third terciles were used to categorize low or high consumption.

#### 2.3.3. Stress-Related Variables

As part of the investigation into the correlates of diabetes, the Kessler 10 scale, a continuous measure of psychological distress, was included. The Kessler 10 consists of 10 items that measure experiences of non-specific anxiety and depressive symptoms in the past 30 days [24]. It has demonstrated adequate psychometric properties for predicting both depression and anxiety in South Africa [25].

Seven indicators of exposure to stress were included in the analysis: hunger-related stress, alcohol-related stress in the household, crowding, neighborhood inaccessibility, economic stress, interpersonal conflict, and crime and alcohol-related stress in the neighborhood. Except for hunger and household crowding, variables for these constructs were created by standardizing and summing the items related to these constructs and then creating indicators for the top quintile of each [26]. Household crowding was operationalized by the number of household members divided by the number of sleeping rooms in one’s house. The hunger-related stress construct was based on the Community Childhood Hunger Identification Project index, which ranges from 0 to 8. Scores from 5 to 8 indicating a high level of household food shortages [27] were used to represent hunger-related stress.

### 2.4. Statistical Analysis

Data were analyzed using Stata version 15.0 (Stata Corporation, College Station, TX, USA). Descriptive statistics were used to summarize characteristics of the study sample by race for the overall sample and by geographic type within the Black South African population. Chi-square tests and ANOVAs were used to compare the differences in categorical variables and continuous variables, respectively. To maintain the power of our analyses, the missing values for all the variables included in the analyses using chained equations were imputed. When correctly implemented, multiple imputation procedures produce asymptotic unbiased estimates and standard errors [28]. A total of 25 imputations were performed for each analysis in this study. A greater number of imputations gives confidence in the replicability of the standard error estimates [29]. Multivariate regression models were performed using the ‘mi: svyset’ command to introduce weights that account for the complex design of the SANHANES survey. The ordinary least squares regression was used to estimate the associations between the explanatory variables and HbA1c. The explanatory variables associated with the presence of diabetes were investigated using logistic regression. Model 1a included the demographic variables race, sex, and age. Model 2a added education, income, and wealth index. Model 3a added psychological distress, and Model 4a added a series of stressor variables. Model 5a added the behavioral and medical risk factors. Each of these models was also run with a ‘b’ counterpart, where the model was applied only to the Black South African population, and geographic type was included along with age and sex in all models.

## 3. Results

### 3.1. Socio-Demographic Characteristics of the Study Sample

The study sample comprised a total of 4598 participants. The majority of the respondents were Black South Africans (66.2%), followed by Coloured respondents (26.7%), Indians (4.9%), and Whites (2.2%) (using the population classifications of Statistics South Africa to classify each of the four race groups). Table 1 presents descriptive statistics for the study sample by race. The mean ages were 40, 50, 41, and 48 years for Black South Africans, Whites, Coloureds, and Indians, respectively. Coloured South Africans had the lowest prevalence of post-high school education (3.4%), followed by Black South Africans (3.9%), Indians (10.7%), and Whites (27.4%). This pattern was similar for per capita household income of greater than R50,000 annually: Coloureds (7.5%), Black South Africans (5.5%), Indian (16.6%), and Whites (37.4%). The household wealth index showed similar gradations, with the percentage of households in the highest quintile being for Black South Africans 10.5%, Coloureds 28.6%, Indians 72.7%, and Whites 82.4%.

### 3.2. Prevalence of Diabetes Mellitus in the General Population

Table 2 shows the weighted prevalence of diabetes and the weighted percent HbA1c by race among South Africans older than 15 years of age. Overall, the weighted prevalence of diabetes was 10.5%. Diabetes prevalence was lowest among Black South Africans (8.9%), followed by Coloureds (9.9%), Whites (16%), and highest among Indians (32.2%). The mean percentage HbA1c score was similar for Black South Africans (5.77%), Whites (5.64%), and Coloureds (5.92%), but higher for Indians (6.33%).

### 3.3. Determinants of Diabetes Mellitus in the General Population

Table 3 shows the multivariate regression model for the national sample with diabetes as the binary outcome variable. In Model 1, compared to Black South Africans, Indians had significantly higher odds of diabetes (AOR = 4.28, *p* < 0.001), and increasing age was associated with higher odds of diabetes (AOR = 1.05, *p* < 0.001). In subsequent models, adjustment for SES, psychological distress, stressors, and risk factors, reduced the racial disparity in the odds of diabetes between Black South Africans and Indians, but the disparity remained significant. After adjustment for the socioeconomic factors in Model 2, diabetes was not significantly associated with the indicators of SES. In Model 3, each unit increase in psychological distress was associated with a 4% increase in the odds of diabetes (AOR = 1.04, *p* = 0.011). In Model 4, with the inclusion of the social stressor variables, the association between psychological distress and diabetes remained significant, with little change to its magnitude (AOR = 1.05, *p* = 0.003). Individuals who had experienced household food shortages had 42% lower odds of diabetes compared to those who had not (AOR = 0.58, *p* = 0.038). Participants who experienced higher crowding also had lower odds of diabetes compared to those who did not (AOR = 0.53, *p* = 0.008).

Finally, with the inclusion of health risk factors in Model 5, Indian race (AOR = 3.35, *p* = 0.006), increasing age (AOR = 1.04, *p* < 0.001), psychological distress (AOR = 1.04, *p* = 0.004), crowding (AOR = 0.51, *p* = 0.006), family history of diabetes (AOR = 2.29, *p* < 0.001), being overweight (AOR = 4.27, *p* = 0.013), and obesity (AOR = 9.99, *p* < 0.001) were significantly associated with diabetes. In this model, experiencing hunger was not significantly associated with the odds of diabetes.

Table 4 presents the results of the multivariate linear regression for the different race groups with HbA1c levels as the outcome variable. In Model 1, compared to Black South Africans, Whites had lower levels (*ß* = –0.36, *p* < 0.001) of HbA1c, and Indians (*ß* = 0.44, *p* = 0.001) had higher levels. In all subsequent models, these racial disparities in HbA1c remained significant. In this, and all subsequent models, age was significantly associated with elevated HbA1c (*ß* = 0.02, *p* = 0.001).

In Model 2, the results showed significant associations between SES and HbA1c. In this model, individuals with higher education had lower HbA1c than those without formal schooling (*ß* = –0.3, *p* = 0.029). Compared to those in the lowest quintile of the household wealth index, those in the fourth quintile (higher SES) had increased HbA1c (*ß* = 0.17, *p* = 0.047). With the inclusion of psychological distress in Model 3, these associations remained significant, with little change to their magnitude. In this model, psychological distress was not significantly associated with HbA1c.

After adding the social stressor variables in Model 4, psychological distress was significantly associated with elevated HbA1c (*ß* = 0.01, *p* = 0.039). The previously described associations between SES and HbA1c remained significant; however, in this model, it was noted that those in the fifth quintile (highest SES) had, on average, higher HbA1c than those in the lowest quintile (*ß* = 0.3, *p* = 0.035). Crowding was the only stressor that was significantly associated with HbA1c (*ß* = –0.11, *p* = 0.012).

Finally, in Model 5, several risk factors, family history of diabetes (*ß* = 0.4, *p* < 0.001), obesity (*ß* = 0.4, *p* < 0.001), and high alcohol use (*ß* = –0.14, *p* = 0.013) were all significantly associated with HbA1c. In addition, age (*ß* = 0.02, *p* < 0.001), higher education (*ß* = –0.28, *p* = 0.037), crowding (*ß* = –0.1, *p* = 0.021), neighborhood crime and alcohol abuse (*ß* = –0.11, *p* = 0.046) were associated with HbA1c. In this final model, Black South Africans continued to have higher HbA1c than Whites (*ß* = –0.49, *p* < 0.001) and lower HbA1c than Indians (*ß* = 0.3, *p* = 0.035).

### 3.4. Correlates of Diabetes in the Black South African Race Group

Further analysis by geographic location was conducted only among Black South Africans. Table 5 displays multivariate models of factors associated with diabetes. In Model 1, there was no variation in the odds of diabetes by geography, with only increasing age (AOR = 1.049, *p* < 0.001) significantly associated with diabetes. In Model 2, none of the SES indicators (education, income, and wealth) were significantly associated with diabetes. In Model 3, psychological distress was significantly associated with diabetes (AOR = 1.038, *p* = 0.031). In Model 4, crowding was the only stressor associated with diabetes with higher levels of crowding significantly associated with lower odds of diabetes, (AOR = 0.51, *p* = 0.016). In addition, psychological distress (AOR = 1.047, *p* = 0.014) remained significantly associated with diabetes. Finally, of the traditional risk factors for diabetes added to Model 5, only obesity (AOR = 7.1, *p* = 0.003) was significantly associated with diabetes. However, age (AOR = 1.05, *p* < 0.001), psychological distress (AOR = 1.046, *p* = 0.01), and crowding (AOR = 0.5, *p* = 0.011) remained associated with diabetes in the final model. Instructively, across all models, neither SES nor geographic location was associated with diabetes.

Table 6 shows the multivariate models of factors associated with elevated HbA1c among Black South Africans by geospatial location. In Models 1 and 2, only age was significantly associated with HbA1c. In Model 3, psychological distress was also unrelated to HbA1c. Of the stressors considered in Model 4, only crowding was significantly associated with HbA1c among Black South Africans (*ß* = –0.12, *p* = 0.021), with higher levels of crowding associated with lower HbA1c levels. In Model 5, family history of diabetes (*ß* = 0.37, *p* < 0.001) and obesity (*ß* = 0.35, *p* < 0.001) were traditional risk factors that had significant positive associations with HbA1c, while high alcohol use had a significant negative association with HbA1c (*ß* = –0.14, *p* = 0.046). In addition, age (*ß* =0.02, *p* < 0.001) and crowding (*ß* = –0.11, *p* = 0.032) remained significantly associated with HbA1c in the final model. Across all models, neither SES nor geographic location was associated with elevated HbA1c.

## 4. Discussion

This study showed that in addition to the well-established risk factors of older age, overweight and obesity, and family history of diabetes, the prevalence of diabetes in South Africa was significantly associated with being of Indian descent, having higher psychological distress, and living in uncrowded households.

Psychological distress as measured by the Kessler 10 scale was one of the few psychosocial factors to show an association with reported diabetes in both the overall sample and the Black South African subsample. More research is needed to investigate the relative influence of different psychosocial stressors on the prevalence of diabetes in South Africa. It is not clear whether psychological distress is a cause of diabetes or whether having diabetes increases psychological distress. The latter hypothesis has more biological plausibility, but further research is needed to elucidate the true pathways of disease causation between psychological distress and the incidence of diabetes. There are several studies supporting a bidirectional association between psychological factors and diabetes [30]. The correlation between diabetes and psychological distress persisted after adjusting for other health risk variables. The finding underscores the importance of dealing with other personal stressors (low self-esteem, emotional disturbances) leading to depression and anxiety, which are common comorbidities among people with diabetes [31]. These findings highlight the importance of dealing with the personal stressors that affect individuals with diabetes by implementing tailored interventions (self-efficacy, coping strategies, social support) that take into account psychological distress related to depression and anxiety [32]. The importance of psychosocial correlates is further highlighted by a European study with a cohort of more than 100,000 participants where the findings show job strain as a risk factor for type 2 diabetes [33]. Job strain was measured to include a wide range of psychosocial aspects such as excessive amounts of work or insufficient time allocated. More research is therefore needed to explore the association of psychosocial correlates and diabetes type 2 but equally important is the direction of this association.

In the overall sample and within the Black South African subsample, household crowding was negatively associated with diabetes prevalence and HbA1c. While counter-intuitive at face value, our measure of household crowding may be conceptualized as a proxy of social support after adjusting for the multitude of negative factors that typically accompany it (low household income, psychological distress, economic stress, hunger, and interpersonal conflict). The pathways by which social support influences diabetes management are well documented in the existing literature. A review by Kadirvelu et al. identified four protective domains of support, all of which have been empirically investigated in South Africa: appraisal—support in the selection of food choices and portion size; information—support in the aggregation of information about diabetes management; instrumental—support in the day-to-day tasks related to food choices and preparation; emotional—support with the psychosocial challenges that accompany living with diabetes [34,35,36].

No statistically significant relationship was found between the other stress indicators and reported diabetes. Part of the reason for this discrepancy may be that while the K10 scale has been validated among low- and middle-income countries, including South Africa [25], the internal reliability and validity of our other proxies for stress are uncertain. In addition, the K10 has been used widely as a screening tool for health-related quality of life [37] and could likely capture, at least in part, the effects of the other stressor variables, such as interpersonal conflict, hunger, and economic or alcohol-related stress. For example, psychological distress, as measured by K10, has been associated with exposure to hunger [38], alcohol consumption-related disorders [39] household financial stress [40], and neighborhood conditions [41]. The construct of other stressor variables may need careful conceptualization and assessment to capture the nuanced and contextualized relevant dimensions of stressful life experiences that are prevalent in South Africa and that may be consequential as risk factors for chronic diseases, such as diabetes.

Diabetes comprises those with elevated blood sugar levels or those currently taking diabetic medication, thereby including the full spectrum of the undiagnosed, the diagnosed and untreated, the treated and controlled, and the treated and uncontrolled. Conversely, lower HbA1c values can reflect diabetes that is controlled by medication or indicates naturally low blood sugar levels. Similarly, higher HbA1c values can reflect both treated uncontrolled diabetes and undiagnosed/untreated diabetes. Various socioeconomic factors are associated with diabetes screening, awareness, treatment adherence, and control, which must be considered when interpreting the variables associated with increased HbA1c levels.

In the overall sample, neighborhood crime and alcohol and high alcohol use showed inverse associations with mean HbA1c levels but not with diabetes. Again, further research is needed to explain this counter-intuitive association. This suggests that after adjusting for other social stressors (including the stress from home alcohol use) and risk factors such as obesity and family history, the lower average HbA1c levels among those in neighborhoods with crime and alcohol use and those who have high alcohol use are explained by other reasons not captured in this study.

No significant associations were found between SES and diabetes. Previous South African studies also found that diabetes prevalence was higher among more affluent socioeconomic groups [42] while others did not [43]. Higher educational level, however, was associated with lower HbA1c, consistent with other studies [44], and possibly reflecting the increased health literacy and access to healthcare that is often associated with higher education.

In addition, there was no association between diabetes (or mean HbA1c levels) and geographic location (urban/rural or formal/informal) in the Black South African subgroup. The null findings regarding SES, stressors, and geographic location should be viewed in light of the changes arising from the epidemiological and nutritional transitions that low and middle-income (LMIC) countries such as South Africa are undergoing. These transitions have resulted in changes in the socio-demographic profile of South Africa, such as increasing education levels, income levels, labor market changes, and mass urban migration patterns in recent years, which may diminish socioeconomic inequalities in diabetes between affluent and previously deprived quintiles or geographic areas as has been observed elsewhere [45]. The socioeconomic changes are initially accompanied by changes in health behaviors toward more calorie-dense foods and decreased physical activity by the wealthier and better-educated early adopters in the population, resulting in an increasing prevalence of NCDs. Later in the epidemiological transition, however, this reverses with diabetes and other NCDs becoming associated with lower socioeconomic quintiles and less educated populations [46], and this was evident in the SANHANES-1 population, where diabetes was less prevalent among those with higher levels of education.

Increasing age was significantly associated with the prevalence of diabetes. Life expectancy at birth has increased by 9 years in South Africa since 2007, presumably due at least in part to the 30% decline in mortality that accompanied the introduction of antiretroviral therapy for HIV/AIDS from 2004 onward [47]. The elderly are more than twice as likely to have diabetes than the middle-aged, with the highest diabetes prevalence observed among 60–74-year-olds [48]. South Africa’s rising life expectancy rates in the past 14 years have therefore led to increased proportions of elderly people who are at risk for NCDs, such as diabetes.

As well as being associated with the rise in life expectancy, this increase in the prevalence and mortality from diabetes in the last 14 years may relate to the large increases in the prevalence of obesity during the past few decades (particularly among South African women) as the adjusted odds ratio for diabetes in the obese was 9.98 in SANHANES-1.

The association between diabetes and family history of diabetes seen in SANHANES-1 can be attributed to genetic effects, as well as to other socio-behavioral risk factors for diabetes that tend to cluster within families and households over time. Risk factors for diabetes, such as being overweight or obese, inactivity, smoking, excessive caloric intake, and poor diet quality [49], are often shared by family members.

The high diabetes prevalence in people of Indian descent has also been observed in many other studies around the world and may relate to risk factors such as increased central obesity and visceral fat, high waist/hip ratio, and hyperinsulinemia [50]. The three-fold higher prevalence of diabetes in the Indian community in South Africa suggests that intensive screening for diabetes is particularly important in this community, beginning in young adulthood. The high prevalence of hypertension in this community [13] makes diabetes screening particularly imperative as these two diseases act synergistically in causing ischemic heart disease and kidney failure [51]. In addition, the high prevalence of smoking in the Indian population [16] multiplicatively increases the risk of heart disease and stroke in South African Indians who also have diabetes and hypertension [52].

Mean HbA1c is an important population metric because the pathological effects of elevated glucose levels occur not only when the HbA1c is above 6.5% but also in those with impaired glucose tolerance (IGT)—HbA1c of 5.5–6.4% and hyperinsulinemia [53]. Hence, health promotion interventions (weight control, diet, exercise, smoking cessation, blood pressure control) should not only be targeted at people with diabetes (HbA1c > 6.5%) but also at those with IGT (HbA1c 4.5–6.4%), and indeed the entire adult population, with particular emphasis on vulnerable communities such as Indians, the elderly, and the obese.

This argues for much greater research efforts into diabetes and its determinants in South Africa (both traditional and non-traditional determinants such as psychosocial factors), as well as for nationwide diabetes screening programs and health promotion interventions, particularly those that address the obesity epidemic in the country.

This study has several limitations that need to be highlighted. Due to the study’s cross-sectional nature, temporal ordering for the relationships among psychosocial factors and other risk factors and diabetes mellitus cannot be inferred. Thus, no claims regarding causality are appropriate. The reliance on self-reports for the our psychosocial variables raises concerns about the validity for a range of reasons, including systematic response distortions, method variance, and the psychometric properties of questionnaire scales. The interpretation of these findings may also be limited by the complex social and behavioral changes attributed to the demographic and epidemiological transitions in South Africa in recent years [54], which cannot be investigated in a cross-sectional study. Another important limitation is the differences in the response rates between men and women. Although we have tried to address this in weighting our data for analysis, it is important to highlight that men are generally less responsive to surveys or accessing healthcare [55]. Nevertheless, the strength of this study is that it is based on a large-scale, nationally representative sample and can be generalized to young people with diabetes and adults 15 years and older in the country. The design of future studies could also benefit from adding measures or data from hospital records where available so there is less reliance on self-reported measures.

## 5. Conclusions

An improved understanding of risk factors related to diabetes can assist in making informed decisions about diabetes programs and policies for improved health outcomes and disease prevention. Diabetes screening programs and health promotion interventions are needed in every community, focusing on risk factors for diabetes such as obesity, poor diet, and lack of physical activity. A particular focus should be made on communities with increased vulnerability to diabetes, such as the Indian community in South Africa, the elderly, those with a family history of diabetes, the overweight/obese, and those with psychological distress. Healthcare planning and delivery are required to improve diabetes screening and access and treatment adherence for those diagnosed among these vulnerable groups. South Africa has implemented community-led public health education initiatives where community health workers screen and deliver health education to households [56], which has the potential to impact not only individuals but whole families.

Health promotion interventions should be designed and implemented at the community level to prevent diabetes (primary prevention) and also to mitigate the effects of the established disease (secondary prevention) in end-organ damage to the heart, kidneys, eyes, brain, circulation, and peripheral nervous system.

## Figures and Tables

**Table 1 ijerph-20-05798-t001:** Characteristics of the study sample comprising youth and adults 15 years and older, South Africa 2012.

Variables	Overall (*n* = 4598)		Black (*n* = 3042)		White (*n* = 103)		Coloured (*n* = 1229)		Indian (*n* = 224)		
	*n* ^a^	Percent ^b^	*n* ^a^	Percent ^b^	*n* ^a^	Percent ^b^	*n* ^a^	Percent ^b^	*n* ^a^	Percent ^b^	*p*
Age < 25	1220	26.5%	876	28.8%	14	13.6%	299	24.3%	31	13.8%	<0.001
Age ≥ 25 and <35	777	16.9%	532	17.5%	10	9.7%	209	17.0%	26	11.6%	
Age ≥ 35 and <45	675	14.7%	428	14.1%	13	12.6%	203	16.5%	31	13.8%	
Age ≥ 45 and <55	755	16.4%	461	15.2%	12	11.7%	236	19.2%	46	20.5%	
Age ≥ 55 and <65	614	13.4%	370	12.2%	34	33.0%	162	13.2%	48	21.4%	
Age ≥ 65	557	12.1%	375	12.3%	20	19.4%	120	9.8%	42	18.8%	
Female	2951	64.2%	1957	64.3%	52	50.5%	798	65.0%	144	64.3%	0.073
Male	1646	35.8%	1085	35.7%	51	49.5%	430	35.0%	80	35.7%	
No formal Schooling/Grade 0–7	1404	35.8%	1010	39.5%	10	10.5%	346	32.8%	38	17.8%	<0.001
Grade 8–12 (or Equivalent)	2334	59.5%	1448	56.6%	59	62.1%	674	63.8%	153	71.5%	
Higher Education	184	4.7%	99	3.9%	26	27.4%	36	3.4%	23	10.7%	
Income < 5000	1321	33.2%	988	37.8%	12	13.2%	255	23.3%	66	37.7%	<0.001
Income ≥ 5000 and <10,000	920	23.2%	637	24.4%	5	5.5%	260	23.7%	18	10.3%	
Income ≥ 10,000 and <25,000	1074	27.0%	639	24.5%	23	25.3%	370	33.8%	42	24.0%	
Income ≥ 25,000 and <50,000	371	9.3%	206	7.9%	17	18.7%	128	11.7%	20	11.4%	
Income ≥ 50,000	288	7.2%	143	5.5%	34	37.4%	82	7.5%	29	16.6%	
Wealth Index Quantile 1 (Low)	759	20.0%	670	26.5%	4	4.7%	84	8.5%	1	0.5%	<0.001
Wealth Index Quantile 2	757	19.9%	617	24.4%	1	1.2%	137	13.8%	2	1.0%	
Wealth Index Quantile 3	758	20.0%	548	21.7%	0	0.0%	189	19.1%	21	10.8%	
Wealth Index Quantile 4	762	20.1%	426	16.9%	10	11.8%	297	30.0%	29	14.9%	
Wealth Index Quantile 5 (High)	759	20.0%	265	10.5%	70	82.4%	283	28.6%	141	72.7%	
Low Hunger	2901	68.9%	1720	61.2%	85	90.4%	892	81.6%	204	95.8%	<0.001
High Hunger	1308	31.1%	1089	38.8%	9	9.6%	201	18.4%	9	4.2%	
Low Home Alcohol Stress	3422	79.9%	2225	77.8%	95	97.9%	902	81.2%	200	93.0%	<0.001
High Home Alcohol Stress	862	20.1%	636	22.2%	2	2.1%	209	18.8%	15	7.0%	
Low Crowding	3189	74.2%	2089	73.1%	93	94.9%	800	71.1%	207	96.3%	<0.001
High Crowding	1107	25.8%	769	26.9%	5	5.1%	325	28.9%	8	3.7%	
Low Neighborhood Inaccessibility	3286	77.0%	2125	74.7%	81	82.7%	875	79.1%	205	94.5%	<0.001
High Neighborhood Inaccessibility	980	23.0%	720	25.3%	17	17.3%	231	20.9%	12	5.5%	
Low Economic Stress	3243	78.2%	2141	76.7%	79	84.0%	864	81.6%	159	78.7%	<0.001
High Economic Stress	902	21.8%	649	23.3%	15	16.0%	195	18.4%	43	21.3%	
Low Interpersonal Conflict	3455	82.2%	2210	79.5%	86	87.8%	989	87.8%	170	85.0%	<0.001
High Interpersonal Conflict	748	17.8%	569	20.5%	12	12.2%	137	12.2%	30	15.0%	
Low Neighborhood Crime and Alcohol	3347	79.4%	2144	76.6%	85	90.4%	927	83.6%	191	89.7%	<0.001
High Neighborhood Crime and Alcohol	869	20.6%	656	23.4%	9	9.6%	182	16.4%	22	10.3%	
No Family History of Diabetes	3011	76.7%	2033	79.4%	63	70.8%	810	74.9%	105	54.1%	<0.001
Family History of Diabetes	913	23.3%	526	20.6%	26	29.2%	272	25.1%	89	45.9%	
Underweight < 18.5 kg/m^2^	377	8.6%	235	8.1%	2	2.1%	120	10.2%	20	9.6%	<0.001
Normal weight 18.5–24.9 kg/m^2^	1811	41.4%	1228	42.4%	26	27.7%	490	41.5%	67	32.2%	
Overweight 25–29.9 kg/m^2^	974	22.3%	623	21.5%	32	34.0%	259	21.9%	60	28.8%	
Obese ≥ 30 kg/m^2^	1215	27.8%	809	27.9%	34	36.2%	311	26.4%	61	29.3%	
Active Lifestyle	1506	37.5%	1026	38.8%	39	41.1%	400	37.0%	41	21.2%	<0.001
Inactive Lifestyle	2506	62.5%	1617	61.2%	56	58.9%	681	63.0%	152	78.8%	
High Fruit/Veg Intake	2805	68.0%	1748	64.1%	83	86.5%	804	73.0%	170	85.9%	<0.001
Low Fruit Veg Intake	1319	32.0%	980	35.9%	13	13.5%	298	27.0%	28	14.1%	
Low Sugar Intake	3458	84.6%	2278	84.3%	83	85.6%	919	84.1%	178	89.9%	0.263
High Sugar Intake	631	15.4%	423	15.7%	14	14.4%	174	15.9%	20	10.1%	
Low Fat Intake	3550	87.6%	2318	86.4%	83	88.3%	965	89.4%	184	92.5%	<0.001
High Fat Intake	504	12.4%	364	13.6%	11	11.7%	114	10.6%	15	7.5%	
Low Alcohol Intake	3623	86.6%	2434	88.3%	86	86.9%	902	80.3%	201	99.5%	<0.001
High Alcohol Intake	559	13.4%	324	11.7%	13	13.1%	221	19.7%	1	0.5%	
Kessler 10 Psychological Distress Scale Score (Mean, SD)	14.28	5.86	14.89	6.15	11.85	3.77	13.28	5.24	12.76	4.33	<0.001

^a^ Unweighted N. ^b^ Weighted %.

**Table 2 ijerph-20-05798-t002:** Weighted Prevalence of diabetes and mean HbA1c by race groups among youth and adults 15 years and older, South Africa.

	Overall (*n* = 4598)		African (*n* = 3042)		White (*n* = 103)		Coloured (*n* = 1229)		Indian (*n* = 224)		
Variables	Percent (S.E.)	95% CI	Percent (S.E.)	95% CI	Percent (S.E.)	95% CI	Percent (S.E.)	95% CI	Percent (S.E.)	95% CI	*p*-Value
Diabetes	10.5 (0.01)	8.3–12.7	8.9 (0.01)	6.5–11.2	16 (0.06)	4.6–27.4	9.9 (0.02)	6.7–13	32.2 (0.07)	19.2–45.1	<0.001
HbA1c	5.79 (0.03)	5.74–5.85	5.77 (0.03)	5.71–5.84	5.64 (0.1)	5.44–5.83	5.92 (0.06)	5.79–6.04	6.33 (0.12)	6.08–6.57	<0.001

**Table 3 ijerph-20-05798-t003:** Multivariate models of factors associated with diabetes mellitus type 2 among youth and adults 15 years and older, South Africa 2012 (***n*** = 4598).

Diabetes	Model 1: Demographic Variables			Model 2: Model 1 + SES Variables			Model 3: Model 2 + Kessler 10			Model 4: Model 3 + Stressors *			Model 5: Model 4 + Risk Factors *		
Predictors	OR	S.E.	*p*	OR	S.E.	*p*	OR	S.E.	*p*	OR	S.E.	*p*	OR	S.E.	*p*
**Population Group**															
Black	*ref*			*ref*			*ref*			*ref*			*ref*		
White	1.226	0.648	0.700	1.009	0.613	0.989	1.144	0.680	0.822	1.033	0.577	0.954	0.883	0.429	0.798
Coloured	1.051	0.275	0.849	0.919	0.255	0.762	1.007	0.263	0.979	0.937	0.252	0.810	1.109	0.288	0.689
Indian	**4.280**	1.710	<0.001	**3.044**	1.347	0.012	**3.374**	1.470	0.006	**2.884**	1.341	0.023	**3.352**	1.476	0.006
**Sex**															
Female	*ref*			*ref*			*ref*			*ref*			*ref*		
Male	0.839	0.242	0.544	0.842	0.235	0.538	0.855	0.241	0.579	0.793	0.202	0.362	1.558	0.456	0.131
**Age in years**	**1.046**	0.008	<0.001	**1.049**	0.007	<0.001	**1.048**	0.007	<0.001	**1.047**	0.007	<0.001	**1.043**	0.007	<0.001
**Educational status**															
No formal Schooling/Grade 0–7				*ref*			*ref*			*ref*			*ref*		
Grade 8–12 (or Equivalent)				1.192	0.288	0.467	1.247	0.311	0.378	1.159	0.275	0.533	1.108	0.270	0.674
Higher Education				0.309	0.232	0.119	0.330	0.248	0.140	0.314	0.231	0.116	0.328	0.209	0.081
**Household income in Rands**															
<5000				*ref*			*ref*			*ref*			*ref*		
≥5000 < 10,000				0.728	0.223	0.301	0.725	0.225	0.300	0.755	0.213	0.319	0.722	0.210	0.264
≥10,000 < 25,000				1.027	0.320	0.933	1.014	0.319	0.966	0.934	0.315	0.840	0.863	0.302	0.674
≥25,000 < 50,000				0.848	0.418	0.739	0.837	0.412	0.718	0.736	0.365	0.537	0.720	0.377	0.531
≥50,000				1.264	0.761	0.698	1.247	0.751	0.713	1.149	0.652	0.806	0.962	0.492	0.939
**Asset-Based Wealth Index**															
Quintile 1 (Lowest SES)				*ref*			*ref*			*ref*			*ref*		
Quintile 2 (Lower SES)				1.763	1.003	0.320	1.825	1.052	0.297	1.640	0.935	0.386	1.491	0.922	0.518
Quintile 3 (Middle SES)				1.387	0.741	0.540	1.398	0.758	0.537	1.267	0.699	0.669	1.034	0.626	0.956
Quintile 4 (Higher SES)				1.720	0.848	0.272	1.743	0.871	0.267	1.511	0.795	0.433	1.168	0.669	0.786
Quintile 5 (Highest SES)				2.445	1.347	0.106	2.551	1.430	0.096	2.188	1.261	0.175	1.410	0.821	0.556
**Kessler10 Score**							**1.038**	0.015	0.011	**1.048**	0.017	0.003	**1.044**	0.015	0.004
**Stressor indicators**															
Hunger										**0.575**	0.153	0.038	0.568	0.168	0.057
Home Alcohol Stress										0.952	0.398	0.905	0.921	0.348	0.828
Crowding										**0.531**	0.126	0.008	**0.510**	0.124	0.006
Neighborhood Inaccessibility										0.789	0.220	0.396	0.810	0.245	0.486
Economic Stress										1.157	0.435	0.698	1.269	0.527	0.567
Conflict										0.718	0.161	0.140	0.704	0.167	0.139
Neighborhood Crime and Alcohol										0.861	0.203	0.526	0.794	0.204	0.371
**Health indicators**															
Family History of Diabetes													**2.288**	0.524	<0.001
Underweight < 18.5 kg/m^2^													*ref*		
Normal weight 18.5–24.9 kg/m^2^													1.862	1.206	0.338
Overweight 25–29.9 kg/m^2^													**4.266**	2.481	0.013
Obese ≥ 30 kg/m^2^													**9.985**	6.005	<0.001
Inactive Lifestyle													1.189	0.303	0.496
Low Fruit and Vegetables													1.263	0.348	0.398
High Sugar													0.803	0.250	0.482
High Fat													1.399	0.502	0.350
High Alcohol Use													0.842	0.398	0.715

**Bold** indicates the estimate is significant at the 0.05 alpha level. * Model also adjusts for alcohol use in home.

**Table 4 ijerph-20-05798-t004:** Multivariate models of factors associated with glycated hemoglobin (HbA1c) among youth and adults 15 years and older, South Africa 2012 (*n* = 4598).

HbA1c	Model 1: Demographic Variables			Model 2: Model 1 + SES Variables			Model 3: Model 2 + Kessler 10			Model 4: Model 3 + Stressors *			Model 5: Model 4 + Risk Factors *		
Predictors	Diff	S.E.	*p*	Diff	S.E.	*p*	Diff	S.E.	*p*	Diff	S.E.	*p*	Diff	S.E.	*p*
**Population Group**															
Black	*Ref*			*ref*			*ref*			*ref*			*ref*		
White	**−0.364**	0.100	<0.001	**−0.412**	0.128	0.001	**−0.388**	0.131	0.003	**−0.436**	0.134	0.001	**−0.485**	0.130	<0.001
Coloured	0.095	0.076	0.214	0.042	0.075	0.580	0.058	0.077	0.449	0.037	0.076	0.628	0.082	0.071	0.253
Indian	**0.437**	0.128	0.001	**0.338**	0.145	0.021	**0.358**	0.147	0.015	**0.326**	0.156	0.037	**0.303**	0.143	0.035
**Sex**															
Female	*Ref*			*ref*			*ref*			*ref*			*ref*		
Male	−0.032	0.054	0.551	−0.029	0.049	0.560	−0.025	0.049	0.607	−0.033	0.048	0.495	0.105	0.054	0.053
**Age in years**	**0.019**	0.002	<0.001	**0.019**	0.002	<0.001	**0.019**	0.002	<0.001	**0.018**	0.002	<0.001	**0.016**	0.002	<0.001
**Educational status**															
No formal Schooling/Grade 0–7				*ref*			*ref*			*ref*			*ref*		
Grade 8–12 (or Equivalent)				−0.008	0.091	0.932	<0.001	0.093	0.996	−0.003	0.092	0.972	−0.014	0.091	0.880
Higher Education				**−0.297**	0.136	0.029	**−0.290**	0.135	0.033	**−0.272**	0.132	0.041	**−0.278**	0.133	0.037
**Household income in Rands**															
<5000				*ref*			*ref*			*ref*			*ref*		
≥5000 < 10,000				0.049	0.078	0.526	0.049	0.078	0.529	0.073	0.077	0.342	0.065	0.074	0.384
≥10,000 < 25,000				−0.009	0.079	0.909	−0.009	0.079	0.908	−0.005	0.079	0.954	−0.027	0.078	0.733
≥25,000 < 50,000				0.010	0.120	0.934	0.005	0.121	0.969	−0.007	0.123	0.956	−0.031	0.119	0.798
≥50,000				0.070	0.132	0.595	0.072	0.132	0.586	0.059	0.132	0.655	0.023	0.127	0.859
**Asset-Based Wealth Index**															
Quintile 1 (Lowest SES)				*ref*			*ref*			*ref*			*ref*		
Quintile 2 (Lower SES)				0.106	0.081	0.190	0.110	0.081	0.174	0.112	0.082	0.174	0.106	0.081	0.192
Quintile 3 (Middle SES)				0.059	0.077	0.449	0.058	0.078	0.457	0.072	0.080	0.369	0.046	0.080	0.565
Quintile 4 (Higher SES)				**0.171**	0.086	0.047	**0.171**	0.086	0.046	**0.195**	0.087	0.026	0.150	0.090	0.098
Quintile 5 (Highest SES)				0.254	0.137	0.064	0.259	0.137	0.059	**0.298**	0.141	0.035	0.205	0.135	0.129
**Kessler10 Score**							0.008	0.005	0.107	**0.011**	0.005	0.039	0.007	0.005	0.133
**Stressor indicators**															
Hunger										0.029	0.051	0.573	0.034	0.052	0.520
Home Alcohol Stress										−0.090	0.076	0.236	−0.088	0.070	0.210
Crowding										**−0.111**	0.044	0.012	**−0.098**	0.042	0.021
Neighborhood Inaccessibility										0.079	0.077	0.307	0.091	0.074	0.221
Economic Stress										−0.010	0.059	0.866	0.001	0.057	0.984
Conflict										−0.105	0.067	0.122	−0.095	0.062	0.128
Neighborhood Crime and Alcohol										−0.096	0.054	0.073	**−0.106**	0.053	0.046
**Health indicators**															
Family History of Diabetes													**0.414**	0.080	<0.001
Underweight < 18.5 kg/m^2^													*ref*		
Normal weight 18.5–24.9 kg/m^2^													<0.001	0.060	0.997
Overweight 25–29.9 kg/m^2^													0.174	0.095	0.070
Obese ≥ 30 kg/m^2^													**0.360**	0.075	<0.001
Inactive Lifestyle													−0.057	0.054	0.289
Low Fruit and Vegetables													0.027	0.059	0.650
High Sugar													0.028	0.079	0.720
High Fat													0.003	0.084	0.973
High Alcohol Use													**−0.142**	0.057	0.013

**Bold** indicates that the estimate is significant at the 0.05 alpha level. * Model also adjusts for alcohol use in the home.

**Table 5 ijerph-20-05798-t005:** Multivariate models of factors associated with diabetes mellitus type 2 among Black South African youth and adults 15 years and older, South Africa 2012 (*n* = 3042).

Diabetes	Model 1: Demographic Variables			Model 2: Model 1 + SES Variables			Model 3: Model 2 + Kessler 10			Model 4: Model 3 + Stressors *			Model 5: Model 4 + Risk Factors *		
Predictors	OR	S.E.	*p*	OR	S.E.	*p*	OR	S.E.	*p*	OR	S.E.	*p*	OR	S.E.	*p*
**Geotype**															
Urban Formal	*Ref*			*ref*			*ref*			*ref*			*ref*		
Urban Informal	0.650	0.198	0.159	0.882	0.289	0.701	0.915	0.297	0.784	1.066	0.327	0.835	1.142	0.376	0.687
Rural Informal (Tribal)	0.935	0.404	0.876	1.312	0.566	0.530	1.392	0.602	0.445	1.608	0.723	0.292	1.588	0.760	0.334
Rural Formal (Farms)	0.663	0.313	0.385	0.933	0.483	0.894	0.971	0.516	0.957	1.286	0.711	0.650	1.452	0.800	0.499
**Sex**															
Female	*Ref*			*ref*			*ref*			*ref*			*ref*		
Male	1.061	0.350	0.858	1.041	0.331	0.899	1.055	0.339	0.868	0.910	0.237	0.717	1.819	0.570	0.057
**Age in years**	**1.049**	0.010	<0.001	**1.055**	0.008	<0.001	**1.053**	0.008	<0.001	**1.054**	0.008	<0.001	**1.050**	0.007	<0.001
**Educational status**															
No formal Schooling/Grade 0–7				*ref*			*ref*			*ref*			*ref*		
Grade 8–12 (or Equivalent)				1.280	0.384	0.412	1.357	0.425	0.331	1.220	0.360	0.501	1.186	0.352	0.567
Higher Education				0.442	0.317	0.257	0.486	0.327	0.285	0.439	0.270	0.183	0.449	0.236	0.128
**Household income in Rands**															
<5000				*ref*			*ref*			*ref*			*ref*		
≥5000 < 10,000				0.727	0.261	0.375	0.727	0.263	0.379	0.770	0.244	0.411	0.725	0.227	0.304
≥10,000 < 25,000				1.120	0.379	0.738	1.115	0.384	0.751	1.044	0.382	0.907	0.998	0.360	0.995
≥25,000 < 50,000				0.892	0.420	0.808	0.863	0.419	0.762	0.777	0.370	0.597	0.696	0.375	0.502
≥50,000				2.164	1.415	0.239	2.159	1.395	0.235	1.904	1.111	0.271	1.433	0.721	0.475
**Asset-Based Wealth Index**															
Quintile 1 (Lowest SES)				*ref*			*ref*			*ref*			*ref*		
Quintile 2 (Lower SES)				1.749	1.088	0.370	1.818	1.139	0.341	1.605	0.966	0.432	1.467	0.920	0.542
Quintile 3 (Middle SES)				1.428	0.909	0.577	1.473	0.944	0.546	1.360	0.837	0.618	1.198	0.784	0.782
Quintile 4 (Higher SES)				1.560	0.962	0.471	1.623	1.006	0.435	1.484	0.906	0.518	1.198	0.806	0.789
Quintile 5 (Highest SES)				2.510	1.727	0.182	2.688	1.876	0.158	2.543	1.745	0.175	1.786	1.276	0.418
**Kessler10 Score**							**1.038**	0.018	0.031	**1.047**	0.020	0.014	**1.046**	0.018	0.010
**Stressor indicators**															
Hunger										0.624	0.187	0.116	0.625	0.204	0.151
Home Alcohol Stress										0.953	0.425	0.915	0.966	0.361	0.925
Crowding										**0.507**	0.142	0.016	**0.501**	0.135	0.011
Neighborhood Inaccessibility										0.566	0.208	0.123	0.547	0.205	0.109
Economic Stress										1.157	0.533	0.752	1.292	0.584	0.572
Conflict										0.636	0.163	0.078	0.640	0.157	0.069
Neighborhood Crime and Alcohol										0.869	0.219	0.577	0.797	0.215	0.400
**Health Indicators**															
Family History of Diabetes													1.600	0.396	0.059
Underweight < 18.5 kg/m^2^													*ref*		
Normal weight 18.5–24.9 kg/m^2^													1.580	1.097	0.511
Overweight 25–29.9 kg/m^2^													3.246	2.042	0.062
Obese ≥ 30 kg/m^2^													**7.090**	4.572	0.003
Inactive Lifestyle													1.004	0.272	0.988
Low Fruit and Vegetables													1.110	0.297	0.698
High Sugar													1.075	0.381	0.838
High Fat													1.057	0.398	0.882
High Alcohol Use													0.523	0.293	0.249

**Bold** indicates that the estimate is significant at the 0.05 alpha level. * Model also adjusts for alcohol use in the home.

**Table 6 ijerph-20-05798-t006:** Multivariate models of factors associated with glycated hemoglobin (HbA1c) among Black South African youth and adults 15 years and older, South Africa 2012 (*n* = 3042).

HbA1c	Model 1: Demographic Variables			Model 2: Model 1 + SES Variables			Model 3: Model 2 + Kessler 10			Model 4: Model 3 + Stressors *			Model 5: Model 4 + Risk Factors *		
Predictors	Diff	S.E.	*p*	Diff	S.E.	*p*	Diff	S.E.	*p*	Diff	S.E.	*p*	Diff	S.E.	*p*
**Geotype**															
Urban Formal	*Ref*			*ref*			*ref*			*ref*			*ref*		
Urban Informal	−0.105	0.067	0.117	−0.033	0.074	0.659	−0.029	0.074	0.701	−0.001	0.073	0.991	0.023	0.075	0.761
Rural Informal (Tribal)	−0.043	0.072	0.550	0.043	0.078	0.586	0.049	0.078	0.531	0.041	0.079	0.608	0.039	0.078	0.619
Rural Formal (Farms)	0.013	0.169	0.937	0.102	0.176	0.562	0.111	0.179	0.537	0.090	0.182	0.623	0.134	0.179	0.454
**Sex**															
Female	*Ref*			*ref*			*ref*			*ref*			*ref*		
Male	−0.021	0.067	0.756	−0.019	0.061	0.763	−0.017	0.061	0.787	−0.034	0.061	0.580	0.117	0.069	0.091
**Age in years**	**0.019**	0.002	<0.001	**0.019**	0.002	<0.001	**0.019**	0.002	<0.001	**0.018**	0.002	<0.001	**0.016**	0.002	<0.001
**Educational status**															
No formal Schooling/Grade 0–7				*ref*			*ref*			*ref*			*ref*		
Grade 8–12 (or Equivalent)				0.010	0.111	0.928	0.017	0.114	0.883	0.001	0.112	0.994	−0.003	0.113	0.976
Higher Education				−0.142	0.168	0.396	−0.138	0.168	0.414	−0.152	0.166	0.360	−0.162	0.173	0.350
**Household income in Rands**															
<5000				*ref*			*ref*			*ref*			*ref*		
≥5000 < 10,000				0.045	0.086	0.602	0.045	0.087	0.605	0.069	0.085	0.415	0.055	0.083	0.507
≥10,000 < 25,000				−0.033	0.093	0.721	−0.033	0.093	0.727	−0.023	0.094	0.807	−0.035	0.092	0.701
≥25,000 < 50,000				0.022	0.153	0.884	0.017	0.156	0.916	0.001	0.158	0.995	−0.033	0.155	0.830
≥50,000				0.139	0.180	0.439	0.142	0.180	0.432	0.125	0.182	0.494	0.063	0.182	0.728
**Asset-Based Wealth Index**															
Quintile 1 (Lowest SES)				*ref*			*ref*			*ref*			*ref*		
Quintile 2 (Lower SES)				0.111	0.086	0.195	0.115	0.085	0.178	0.101	0.088	0.252	0.099	0.086	0.253
Quintile 3 (Middle SES)				0.063	0.089	0.476	0.066	0.088	0.458	0.059	0.088	0.509	0.058	0.087	0.510
Quintile 4 (Higher SES)				0.152	0.100	0.128	0.156	0.099	0.116	0.157	0.101	0.121	0.132	0.105	0.206
Quintile 5 (Highest SES)				0.268	0.182	0.141	0.276	0.180	0.127	0.303	0.180	0.093	0.253	0.178	0.157
**Kessler10 Score**							0.006	0.006	0.304	0.007	0.006	0.187	0.005	0.005	0.290
**Stressor indicators**															
Hunger										0.029	0.053	0.589	0.033	0.054	0.542
Home Alcohol Stress										−0.100	0.089	0.263	−0.093	0.082	0.259
Crowding										**−0.119**	0.052	0.021	**−0.105**	0.049	0.032
Neighborhood Inaccessibility										−0.006	0.085	0.945	−0.009	0.081	0.907
Economic Stress										−0.004	0.067	0.952	0.010	0.065	0.877
Conflict										−0.103	0.080	0.196	−0.089	0.075	0.234
Neighborhood Crime and Alcohol										−0.093	0.063	0.137	−0.103	0.061	0.096
**Health indicators**															
Family History of Diabetes													**0.373**	0.096	<0.001
Underweight < 18.5 kg/m^2^													*ref*		
Normal weight 18.5–24.9 kg/m^2^													−0.022	0.074	0.763
Overweight 25–29.9 kg/m^2^													0.108	0.120	0.369
Obese ≥ 30 kg/m^2^													**0.345**	0.098	<0.001
Inactive Lifestyle													−0.046	0.063	0.466
Low Fruit and Vegetables													0.058	0.072	0.414
High Sugar													−0.003	0.091	0.978
High Fat													0.002	0.094	0.986
High Alcohol Use													**−0.141**	0.071	0.046

**Bold** indicates that the estimate is significant at the 0.05 alpha level. * Model also adjusts for alcohol use in the home.

## Data Availability

Data and materials are available from the lead author upon reasonable request.

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
