# Peer review of "Prevalence and Psychosocial Correlates of Diabetes Mellitus in South Africa: Results from the South African National Health and Nutrition Examination Survey (SANHANES-1)"

_ijerph, 2023, doi:10.3390/ijerph20105798_

Round 1
Reviewer 1 Report
This is an interesting and well-written report on diabetes epidemiology in South Africa.
It is valuable for Western readerships to be reminded that there are countries which have hunger and crowdedness problems.
Methodologically I reacted when I realized that there is an enormous difference in the participation rate among men and women in this household-based study. It is only in the white racial groups that the participation rates among men and women are the same. The participation rate among women is twice as high among women than among men in the African, coloured and Indian groups. This could of course introduce substantial bias problems and I cannot see that the authors discuss this problem. It would be interesting for the readers to see results from separate male and female analyses. I do not feel that the use of sex as a confounder solves this problem since differences in correlational patterns could exist in sex-separated analyses.
We need a discussion regarding why men in all the non-white groups do not participate to the same extent as women and how this unwillingness could potentially affect the results. Or is it unwillingness? For the uninformed reader this is unknown territory. Are there actually fewer men in these non-white households and if so how did that happen?
I think the discussion regarding societies in transition is quite interesting, as well as the discussion regarding stress and diabetes. However, the authors make an un-referenced remark regarding the possible bidirectional association between stress and diabetes – that a reversed direction (diabetes causing stress) is more plausible than a forward association (stress increasing likelihood of developing diabetes 2). There is increasing epidemiological evidence from prospective studies showing a forward association, for instance the European IPDWork study which comprises 100 000 participants (Nyberg et al 2014, Diabetes Care).
The formal aspects are satisfactory. The language is very good and the statistical analyses are well executed and described. One general comment is of course that the tables are very voluminous and not so easy to read. But i guess that is inevitable.
Reviewer 2 Report
Thank you for the opportunity to review this paper. The paper is very interesting evaluating the impact of socioeconomic factors and psychosocial stress to diabetes mellitus in South Africa.
The paper is overall very well written and presents an interesting narrative. The use of HbA1c and Diabetes medication as outcomes is a great idea that presents a more complete picture. I particularly like how the model was re-evaluated again within the Black African portion of the sample. This is important since a larger proportion of the population lies within this group and is this group as well the one suffering the heaviest from disparities.
This paper looks great, I would be happy to endorse its publication provided that some clarification or additional input is provided for some comments/concerns I have.
Main Comments/Concerns:
Even when significant on some occasions, psychosocial stress does not appear to be a major risk factor above socioeconomic factors. The title is a bit misleading in that sense, making it sound like it is the major finding. I would recommend changing the title to something more accurate of the findings.
The data is old (2012), this may be the latest, but I wonder if there would be any evidence to support that the findings presented are still relevant at this time. The psychosocial aspect may have been greatly affected after the pandemic.
The inclusion of participants that are 15 is a bit unconventional. 18 is the legal age in South Africa if I understand correctly. These participants are still considered pediatric, less likely to have type 2 diabetes, should they be included in the sample?
What about the population that has above 6.5% HbA1c but is not taking medication? This would be an immediately vulnerable population subset, are there any special characteristics/associations within this group. Maybe doing something similar to the black African re-evaluation.
Minor comments/concerns:
In table 1, the ranges in age, and income should be inclusive of the limit value on one side of the scale using the ≤ or ≥ symbols.
Were there any concerns with normality? just wondering since this data uses a lot of percentage data that may be distributed weirdly.
A lot of the variables are correlated, any issues with collinearity?
The sub header 4.1 appears to be cut.
Author Response
Review Report (Reviewer 2)
Dear Reviewer,
We trust that you are well. We would like to take this opportunity to thank you for reading and commenting on our manuscript. We have incorporated your insightful comments and suggestions into our paper. The paper is much improved thank you.
Here below our responses (in italic) to your specific comments.
Comments and Suggestions for Authors
Thank you for the opportunity to review this paper. The paper is very interesting evaluating the impact of socioeconomic factors and psychosocial stress to diabetes mellitus in South Africa.
The paper is overall very well written and presents an interesting narrative. The use of HbA1c and Diabetes medication as outcomes is a great idea that presents a more complete picture. I particularly like how the model was re-evaluated again within the Black African portion of the sample. This is important since a larger proportion of the population lies within this group and is this group as well the one suffering the heaviest from disparities.
This paper looks great, I would be happy to endorse its publication provided that some clarification or additional input is provided for some comments/concerns I have.
Main Comments/Concerns:
Even when significant on some occasions, psychosocial stress does not appear to be a major risk factor above socioeconomic factors. The title is a bit misleading in that sense, making it sound like it is the major finding. I would recommend changing the title to something more accurate of the findings.
Response:
Thank you for the comments.
We were intentional in deciding the title to reflect “correlates” which is inclusive of the “stress”. The paper set out to investigate these correlates and even though there was minimal association in our findings we still feel that this research into the non-traditional determinants of diabetes warrant more focused research. This is why we feel that the title in the current form should stand.
The data is old (2012), this may be the latest, but I wonder if there would be any evidence to support that the findings presented are still relevant at this time. The psychosocial aspect may have been greatly affected after the pandemic.
Response:
Thank you for the comments
We emphatically agree with the reviewers comments that the psychosocial aspects may have been greatly affected by the pandemic in more recent times. The data is old but it was the last time a nationally representative study of this nature was conducted.
The inclusion of participants that are 15 is a bit unconventional. 18 is the legal age in South Africa if I understand correctly. These participants are still considered pediatric, less likely to have type 2 diabetes, should they be included in the sample?
Response:
Thank you for the comments.
Part of our objectives was to also investigate the prevalence of obesity as risk factor for diabetes type 2. 15 years olds could be obese and they at risk for diabetes so they were included for those reasons.
What about the population that has above 6.5% HbA1c but is not taking medication? This would be an immediately vulnerable population subset, are there any special characteristics/associations within this group. Maybe doing something similar to the black African re-evaluation.
Response:
Thank you for the comments.
Yes, we agree with the reviewer that this population would be a vulnerable subset. This was however not the focus for this study. Moreover, other reviewers have suggested that the paper as written is too long, so we prefer not to conduct additional analyses that may distract from the main message of the paper.
Minor comments/concerns:
In table 1, the ranges in age, and income should be inclusive of the limit value on one side of the scale using the ≤ or ≥ symbols.
Thank you for this attention to detail. We have made the change.
Were there any concerns with normality? just wondering since this data uses a lot of percentage data that may be distributed weirdly.
Given our large sample size, we did not have concerns with the violation of the normality assumption[1]. Moreover, although log transformations in the presence of non-normality are common, we chose to not log transform our HbA1c values to ensure that our results were interpretable.
A lot of the variables are correlated, any issues with collinearity?
Thank you for this comment – we assessed the presence of collinearity with a correlation matrix of the variables included in our models. Please see attached – with the exception of HbA1c and Diabetes, no variables had a correlation at or above 0.7[2].
The sub header 4.1 appears to be cut.
Response:
Thank you for the comments.
There is no sub header 4.1, this was a formatting error which has been corrected. (Line – 459)
[1] Ghasemi A, Zahediasl S. Normality tests for statistical analysis: a guide for non-statisticians. Int J Endocrinol Metab. 2012 Spring;10(2):486-9. doi: 10.5812/ijem.3505. Epub 2012 Apr 20. PMID: 23843808; PMCID: PMC3693611.
[2] Belinda, B. and Peat, J., Medical statistics: a guide to SPSS, data analysis, and critical appraisal (2nd edition), Wiley, UK, 2014.

Reviewer 3 Report
General comments.
The scope of the article is on the description and estimation of the prevalence of diabetes mellitus in a wide range of its correlates in South Africa, with a strong emphasis on race and psychosocial aspects. The last can be considered as the main novelty of the study. Data for the analysis are coming from the National Health Survey in 2012, therefore, could be considered as outdated, because diabetes prevalence in South Africa is changing dramatically according to the data mentioned in the article (line 55). However, the relationship with the set of predictors could be the same to some degree. Could it be discussed?
The article takes 20 pages, which is too much for the research paper. (Tables or independent variable with no statistical significance to appendix?).
The prevalence of diabetes is assessed by self-reported diagnosis and/or HbA1c level criterion. The latter is reflecting not only the presence of known or unknown DM, but also a level of its control, which makes additional difficulties in interpreting results, particularly in relation to psychosocial factors (e.g., could be differences for known and unknown diabetes). A limitation?
Conclusions are not the direct outcome of the results and can be drown without this study. It is possible to rewrite conclusions to become more specific in relation to the results obtained and their interpretation particularly with respect to psychosocial factors, as this is the objective of the study.
Special comments.
1. The section 'Introduction'
· Line 114: unclear terms: 'formal' and 'informal'. Only brackets in Table 5 for the first time allow to guess that this is about 'Tribal' and 'Farms'.
· Line 127: can lead the reader to think about geographic locations like regions, counties, municipalities, etc.
2. The section 'Materials and methods':
· Line 161: It is not clear for the international reader the meaning of 'Coloured' (difference from black African) and 'Indian'. Indian is a heterogeneous ethnicity (or nationality?) and not a race. Are Indians recent immigrants and why are their social parameters higher (table 1)?
3. The section 'Results': no special comments
4. The section 'Discussion':
· Line 404: it is difficult to imagine these 'other reasons'. Would it be possible to add hypothesis?
· Lines 457 to 464: can be omitted, not related to the results.
5. Section 'Conclusions': see general comments.
